# Thiocoumarin Caged Nucleotides: Synthetic Access and Their Photophysical Properties

**DOI:** 10.3390/molecules25225325

**Published:** 2020-11-15

**Authors:** Jiahui Ma, Alexander Ripp, Daniel Wassy, Tobias Dürr, Danye Qiu, Markus Häner, Thomas Haas, Christoph Popp, Dominik Bezold, Sabine Richert, Birgit Esser, Henning J. Jessen

**Affiliations:** 1Institute of Organic Chemistry, University of Freiburg, Albertstr. 21, 79104 Freiburg, Germany; jiahui.ma@oc.uni-freiburg.de (J.M.); alexander.ripp@livmats.uni-freiburg.de (A.R.); daniel.wassy@oc.uni-freiburg.de (D.W.); tobias.duerr@oc.uni-freiburg.de (T.D.); danyeqiu@gmail.com (D.Q.); markus.haener@oc.uni-freiburg.de (M.H.); thomas.haas@oc.uni-freiburg.de (T.H.); christoph.popp@ocbc.uni-freiburg.de (C.P.); dominik.bezold@ocbc.uni-freiburg.de (D.B.); besser@oc.uni-freiburg.de (B.E.); 2Cluster of Excellence livMatS @ FIT—Freiburg Center for Interactive Materials and Bioinspired Technologies, University of Freiburg, Georges-Köhler-Allee 105, 79110 Freiburg, Germany; 3Institute of Physical Chemistry, University of Freiburg, Albertstr. 21, 79104 Freiburg, Germany; sabine.richert@physchem.uni-freiburg.de

**Keywords:** thio-DEACM, photocage, nucleotides, photochemistry

## Abstract

Photocages have been successfully applied in cellular signaling studies for the controlled release of metabolites with high spatio-temporal resolution. Commonly, coumarin photocages are activated by UV light and the quantum yields of uncaging are relatively low, which can limit their applications in vivo. Here, syntheses, the determination of the photophysical properties, and quantum chemical calculations of 7-diethylamino-4-hydroxymethyl-thiocoumarin (thio-DEACM) and caged adenine nucleotides are reported and compared to the widely used 7-diethylamino-4-hydroxymethyl-coumarin (DEACM) caging group. In this comparison, thio-DEACM stands out as a phosphate cage with improved photophysical properties, such as red-shifted absorption and significantly faster photolysis kinetics.

## 1. Introduction

Chemists had started to use light as an external trigger around the 1970s to control the release of various molecules [1,2]. Light triggers relying on photocleavable protecting groups (PPGs or photocages) have shown significant utility for the study and control of biological processes in living cells and tissues [1]. Among the numerous available photocages, one of the most widely applied classes is based on the coumarin scaffold. Coumarin derivatives exhibit a range of appealing benefits as caging groups, for example, easy synthesis, high biocompatibility [3], flexibility of structural modifications [4] with tunable spectral properties [5], while the mechanism of photocleavage is well understood [6,7,8]. Attractive applications of coumarin derivatives in chemical biology include cell signaling molecules [4,9,10,11], neurotransmitters [12,13,14,15], peptides [16], and photosensitizers [17]. Furthermore, additional photocages based on the coumarin scaffold are being developed, such as derivatives with modifications at the 2-position [18,19,20,21], azetidinyl substituents at the 7-position [22], π-extension at the 3-position [5], and replacement of the oxygen with sulfur on the lactone carbonyl group [23].

In particular, photocaging of nucleotides as cellular signals is an important area of research. Adenosine 5’-triphosphate (ATP) is not only the essential intracellular energy source, but is also involved in many physiological processes [24]. Not surprisingly, different photocages were installed on ATP, for example, 2-nitrobenzyl (commercially available) [2], l-(2-nitro)phenylethyl [2], P3-3’,5’-dimethoxybenzoin ester (DMB) [25,26], p-hydroxyphenacyl (pHP) [27], [7-(dimethylamino)coumarin-4-yl]methyl (DMACM) [28], and [7-(diethylamino)coumarin-4-yl]methyl (DEACM) [29]. Most of these caged ATP molecules are activated by UV light, which impaired their applications in cells and tissue, apart from limited cellular uptake.

Thionated coumarins were first reported in 2012 [30] as photocages for carboxylates (Figure 1). Compared to the corresponding carbonyl precursors, they show a considerable bathochromic shift (between 47 and 73 nm) in absorption and faster uncaging kinetics upon irradiation. In 2013, Jullien’s group also used the strategy to replace oxygen by sulfur, this time on the well-known DEACM photocage [31]. The red-shift in absorption was further increased because of the strong electron donating Et_2_N group in the 7-position. Additionally, in their study a carboxylate was released (benzoate), and blue light was used to trigger the reaction. 7-diethylamino-4-hydroxymethyl-thiocoumarin (thio-DEACM) was also used to cage a cyclofen-OH analogue (Figure 1, tcInd) as a carbamate and applied in living zebrafish embryos [23]. This was the first report where thio-DEACM was applied in an in vivo study. In 2015, benzoic acid-protected thio-DEACM was also used in upconverting nanoparticles (UCNPs)-assisted photochemistry [32]. In 2019, thio-DEACM was introduced to trimethoprim (TMP) [33], again for the caging of carbamates. After uncaging with low-intensity light, TMP was able to activate Cas9. Temporal and spatial control over Cas9 activity by photocaged TMP was demonstrated, and importantly the photolysis was nontoxic to both mammalian and bacterial cells.

These reports underline the attractive features of thio-DEACM as caging group, such as rapid blue-cyan light responsiveness and avoiding UV irradiation of cells in combination with the absence of toxicity of the released photocage. However, unlike the widely used nitrobenzyl (NB) photocage or several coumarin derivatives, particularly DEACM, thio-DEACM has received only little attention. This may be due to impediments regarding its synthesis and also its incorporation into target structures, particularly phosphoric anhydrides, which have not been caged with thio-DEACM yet. To evaluate the utility of thio-DEACM for caging of phosphoric anhydrides, we synthesized several thio-DEACM-caged adenine nucleotides and compared them to their DEACM-caged siblings. We found that orthogonal uncaging of the different photocages with visible light is possible (Scheme 1), which will enable applications of thio-DEACM-caged phosphates in cell and tissue studies in the future.

## 2. Results

### 2.1. Chemistry

DEACM alcohol **4** was synthesized over three steps in 56% overall yield from commercially available starting materials following a previously reported procedure (Appendix A) [34]. Afterwards, thio-DEACM was obtained by replacing the carbonyl oxygen with sulfur using Lawesson’s reagent [30] in 65% yield (Scheme 2A). Subsequently, thio-DEACM was incorporated into phosphorodiamidite **8** [35] (Scheme 2B) giving phosphoramidite (P-amidite) **9**, which serves as a general precursor for phosphoric anhydride synthesis following previously published P(III)-P(V) anhydride synthesis protocols [36]. Overall, P-amidite **9** was obtained in eight steps and 12% yield.

In brief, P-amidite **9** was utilized in the synthesis of modified phosphoric anhydrides generated on unprotected nucleotides (here: AMP, ADP, ATP) via a coupling, oxidation and deprotection strategy [36]. 5-(Ethylthio)-1H-tetrazole (ETT) was used as an activator for the coupling step, giving P(III)-P(V) intermediates **10** (Scheme 2C). In the same flask, *meta*-chloroperbenzoic acid (*m*CPBA) was used for oxidation to a P(V)-P(V) anhydride **11**. After complete oxidation, piperidine was added to remove the Fm group. This represents a convenient synthesis, as all three steps can be conducted in one flask and proceed rapidly. The crude products **12**–**14** were precipitated as sodium salts by addition of a 0.5 M NaClO_4_ solution in acetone [37,38] and purified by strong anion-exchange (SAX) chromatography with an NH_4_HCO_3_—buffer gradient. The product containing fractions were combined, lyophilized, and then again purified by reverse phase chromatography (RP-C18AQ) with acetonitrile in H_2_O with triethylammonium acetate (TEAA) buffer as eluent. We observed formation of DEACM-caged ADP (**15**), ATP (**16**) and AP_4_ (**17**) as minor impurities. Apparently, during the oxidation step with *m*CPBA, thio-DEACM was partially oxidized to DEACM. After purification, DEACM- and thio-DEACM-caged nucleotides **12**–**17** were obtained. Even though the product nucleotides only differ in one atom (O vs. S on DEACM or thio-DEACM), they were readily separated by preparative SAX chromatography. In this regard, the minor side-reaction was useful to also obtain DEACM-caged nucleotides **15**–**17** for direct comparison with thio-DEACM-modified analogues **12**–**14** as discussed in the next section.

### 2.2. Photophysical Properties

To compare the photophysical properties of DEACM **4**, thio-DEACM **7** and their corresponding caged phosphate esters **12**–**14** and **15**–**17**, absorption maxima ($\lambda_{abs}^{max}$), fluorescence maxima ($\lambda_{fl}^{max}$), molar absorption coefficients *ε* ($\lambda_{abs}^{max}$), and fluorescence quantum yields (*Φ*_fl_) were measured. Moreover, density functional theory (DFT) calculations were performed to predict absorption maxima of the alcohols, which are in good agreement with the experimental values. The values are listed in Table 1 (Experimental details, Appendix A).

Thio-DEACM **7** and thio-DEACM-caged nucleotides **12**–**14** showed a bathochromic shift in absorption and emission compared to DEACM, as expected [31]. Additionally, the fluorescence quantum yields are significantly reduced. As the mechanism of photocleavage requires heterolytic bond scission in the excited state, radiative deactivation by fluorescence impairs this process, as do other non-radiative deactivation processes [7]. The lower deactivation by fluorescence, as found for the thio-DEACM cage, boded well for increased quantum yields of uncaging, even though the dark deactivation pathways are not known.

To provide a first theoretical explanation for the experimental results regarding the bathochromic shifts and to explore additional interesting modifications on the coumarin backbone, density functional theory (DFT) calculations were conducted (Figure 2). We also included a selenium substituted coumarin in the calculations and predict a further red-shift in comparison to DEACM and thio-DEACM.

Kim and Lee previously investigated the heavy atom effect of selenium in phenoselazines [39]. They screened several density functionals with varying Fock exchange and found B3LYP to best reproduce the experimental spectra. Thus, the B3LYP [40,41] hybrid functional with the def2-TZVPP [42,43] basis set was used in our studies. A relatively large basis set with extra polarization functions is needed to describe the rather diffuse electrons of selenium and sulfur. The Polarizable Continuum Model (PCM) for water, as implemented in Gaussian16 was used in all calculations.

The values represented in Figure 2 show a clear trend in the orbital energies from oxygen to selenium. The energy of the lowest unoccupied molecular orbitals (LUMOs) decreases from −1.90 (oxygen) to −2.57 eV (selenium), whereas the energies of the highest occupied molecular orbitals (HOMOs) remain the same within the chalcogens (around −5.60 eV). This leads to a decrease in the energy gap from 3.67 (oxygen) to 3.02 eV (selenium).

Time dependent (TD)-DFT calculations provided absorption spectra of the photocages (Appendix A). The first 15 singlet excitations were calculated on the TD-B3LYP/def2-TZVPP level of theory. The resulting longest wavelength absorption maxima are 374 nm (O), 435 nm (S) and 457 nm (Se). These are all attributed to HOMO to LUMO excitations. The same trend observed in the frontier molecular orbital energies was also found here, namely a decreasing band gap from O to Se and a red-shift in absorption.

### 2.3. Photolysis Studies

The stability of caged molecules in water in the dark is very important in photolysis studies and furthermore in biological experiments. This is particularly the case for potentially labile phosphoric anhydrides as present in the caged nucleotides **12**–**14**. Thus, the stability of thio-DEACM-caged nucleotides in the dark at different concentrations (0.1 mM, 0.5 mM, 1 mM) was evaluated over a period of 24 h by HPLC-MS. All of the analyzed molecules were found to be stable (Appendix A).

Next, uncaging experiments were performed using a commercially available LED setup with two distinct wavelengths (Mightex^®^ High-Power LED Collimator Sources, Martinsried, Germany; for the 400 nm LED: 48 mm aperture, typical output power 265 mW; for the 490 nm LED: 22 mm aperture, typical output power 140 mW). The emission spectra of the LEDs were measured with an (Ocean Insight^®^ USB 2000+, Ostfildern, Germany) miniature spectrometer (Figure 3a), showing narrow emission bands with no significant overlap. The actual center wavelength for the 400 nm LED is 396 nm, and full width at half maximum (FWHM) is 13 nm. The actual center wavelength for the 490 nm LED is 494 nm, and FWHM is 29 nm. The LED emissions match well with the absorption spectra of the two different photocaged molecules (Figure 3).

Subsequently, 1 mM sample solutions of the caged nucleotides were prepared in ultrapure water in glass vials and placed 3 cm away from the light source (Experimental setup, Appendix A). Irradiation with different light intensities and for different time intervals were done as reported below. Afterwards, the samples were directly analyzed by HPLC-UV (Thermo, Darmstadt, Germany). The percentage of remaining caged compound was calculated as well as the uncaging ratio. Released nucleotides were plotted according to the uncaging ratios. (Appendix A).

The uncaging kinetics of thio-DEACM-modified ATP **13** were measured at different wavelengths, light intensities, and time points and compared to DEACM-caged ATP **16**. At 490 nm, three different light intensities were used for the photolysis experiments. Thio-DEACM-caged ATP is converted to ATP at high rates, even with 10% of the nominal light intensity. The process was also monitored by ^31^P NMR spectroscopy (Figure 4 and Appendix A). From NMR monitoring, thio-DEACM-caged ATP **13** was cleanly uncaged upon irradiation, and ATP was the only phosphorous-containing product. Since the sensitivity of NMR spectroscopy is limited, capillary electrophoresis (CE)—MS analysis (Appendix A) was also applied to identify the released ATP.

To study differences in the performance of DEACM- and thio-DEACM-caged molecules in response to irradiation at different wavelengths and intensities, the photolysis of DEACM-caged ATP **16** was evaluated in parallel. At 490 nm, no removal of the DEACM cage was observed (Appendix A), since DEACM-caged ATP **16** does not absorb light at this wavelength (Figure 3b, black line). Thus, only thio-DEACM-caged ATP **13** can be uncaged with 490 nm light (Figure 5a).

To photolyze DEACM-caged ATP **16**, a 400 nm LED was used, which emits close to its absorption maximum (Figure 3). With 400 nm light irradiation, the photolysis of **16** was complete in ca. 18 min using 50% or 100% of the nominal LED light intensity at 1 mM concentration (Figure 5b, black and red lines). The same conditions applied to thio-DEACM-caged ATP **13** (50% light intensity, 400 nm) led to complete ATP release within 30 s (Figure 5b, blue line), despite the fact that thio-DEACM-caged ATP has lower absorbance at this wavelength (Table 1).

The same behavior was observed also for caged ADPs **12** and **15** (Figure 6a) and caged AP_4_s [44] **14** and **17** (Figure 6b). This underscores the remarkable utility of thio-DEACM as a phosphate caging group.

Overall, the vastly different reaction rates and light responsiveness indicated that DEACM- and thio-DEACM-caged molecules can be uncaged orthogonally. A mixture of DEACM- and thio-DEACM-caged ATP **13** and **16** proved this selectivity, as only **13** was cleaved upon irradiation at 490 nm (Figure 7a). After full release from thio-DEACM-caged ATP **13**, irradiation with the 400 nm LED released additional ATP from DEACM-caged ATP **16**. This process was monitored by HPLC-UV (Figure 7a). Again, it is noteworthy that despite the fact that the compounds only differ in a single atom (S vs. O), baseline separation was readily achieved using reversed phase HPLC. Likewise, the orthogonal uncaging could also be conducted with thio-DEACM- and DEACM-caged ADP **12** and **15** (Figure 7b) and AP_4_
**14** and **17** (Figure 7c).

Furthermore, when DEACM-caged ATP **16** and thio-DEACM-caged ADP **12** were mixed, 490 nm light can selectively liberate ADP, followed by release of ATP at 400 nm (Figure 8a). The DEACM and thio-DEACM photocages enable a very clean orthogonal uncaging at the selected wavelengths. Successive and clean uncaging of DEACM-caged AP_4_
**17** and thio-DEACM-caged ADP **12**, DEACM-caged AP_4_
**17** and thio-DEACM-caged ATP **13** were also possible (Figure 8b,c).

From the photolysis studies, it is apparent that the formation of DEACM alcohol **4** can also be monitored. However, thio-DEACM **7** was not visible in the HPLC-UV analyses. We speculated that this is due to its low solubility in water. Therefore, aliquots from uncaging experiments were extracted with DCM (in which thio-DEACM alcohol **7** is soluble) after irradiation and analyzed by HPLC-UV. Significant amounts of thio-DEACM **7** were identified (Appendix A).

## 3. Discussion

Thio-DEACM is a recent addition to the family of coumarin-derived photocages. It has previously been used to cage carboxylic acids [30,31,32] and carbamates as amine precursors [23,33]. However, it had not been introduced to phosphates. Here, we disclosed a synthetic procedure based on P-amidite chemistry to introduce the thio-DEACM photocage to adenosine oligophosphates (ADP, ATP, AP_4_). This procedure will also be useful to introduce the thio-DEACM photocage to any phosphoric anhydride based on the modularity of the approach. We obtained small quantities of the DEACM-caged nucleotides as well, which could be readily separated by chromatography. This enabled us to directly compare the performance of the widely used DEACM photocage with thio-DEACM. We found that thio-DEACM has several advantages in uncaging experiments compared to DEACM. First, there is a significant red-shift in the UV-Vis absorption of about 90 nm, also predicted by our DFT calculations. Moreover, we find a largely reduced fluorescence quantum yield of thio-DEACM, which translated into a largely increased photouncaging efficiency. Importantly, thio-DEACM can be removed with commercial LEDs both at 490 and 400 nm, while DEACM is only cleaved at 400 nm. Even at this wavelength, the thio-DEACM photocage is cleaved much more rapidly than its oxygen-containing sibling. We demonstrated the utility of this behavior by selective uncaging of nucleotides mixtures. We were able to conduct the uncaging experiments at relatively high concentrations (up to 1 mM) without significant interference of cleavage products. Thio-DEACM is therefore an attractive choice for caging of phosphates when compared to other caging groups regularly used in combination with phosphates [9,44].

## 4. Conclusions

The quest for the development of red-shifted photocages is important to provide better compatibility with biological systems [45,46,47,48] both in terms of toxicity and penetration depth of the incident light in tissues. The possibility of low-intensity light irradiation, combined with visible light activation and rapid uncaging render thio-DEACM an alternative to the widely used DEACM, when caging of phosphates is required. By substitution of a single atom from O to S, the coumarin scaffold is largely improved with regards to photocleavage properties, while retaining its stability in aqueous solution when attached to phosphoric anhydrides. Moreover, thio-DEACM shows no toxicity in live animals after photocleavage as already demonstrated in previous studies [23]. The chemistry that has been introduced here is highly modular and will help to install thio-DEACM on a range of phosphate-containing signaling molecules that can then be studied in live cell experiments.

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
