# Peer review of "Thiocoumarin Caged Nucleotides: Synthetic Access and Their Photophysical Properties"

_molecules, 2020, doi:10.3390/molecules25225325_

Round 1
Reviewer 1 Report
The manuscript describes synthetic approach to novel thiocoumarin derivatives conjugated with adenine nucleotides, the determination of their photophysical properties and photolysis studies of compounds using LED source. For comparison, coumarin analogs were also synthesized and investigated. Obtained thiocoumarin photocages possess valuable characteristic for future live cell experiments. This work is thorough, carefully planned and meticulously carried out. The manuscript undoubtedly deserves publication in Molecules, but some comments came up when reading:
1) The experimental photophysical characteristics for coumarin and thiocoumarin were obtained for methanol solution, but DFT calculations were performed for water as a solvent. Could the authors comment on the choice of solvent for DFT calculations?
2) Is there any difference in the reactivity of previously reported thiocoumarin carboxylates derivatives under photolysis compared to phosphate esters?
3) Cartesian coordinates and thermochemistry data should be given in SI for all optimized structures.
Author Response
1) The experimental photophysical characteristics for coumarin and thiocoumarin were obtained for methanol solution, but DFT calculations were performed for water as a solvent. Could the authors comment on the choice of solvent for DFT calculations?
The solubility of thiocoumarin in particular prevented us from getting good experimenteal data in water. We have therefore switched to methanol. However, as the molecules will be used in an aqueous environment, we are more interested in the properties in water. To strengthen this point, we have added more data to our table 1 and also exchanged part b in figure 3. There is a slight bathochromic shift in absorption if one moves from methanol to water as solvent.
2) Is there any difference in the reactivity of previously reported thiocoumarin carboxylates derivatives under photolysis compared to phosphate esters?
Yes, the modified phosphates are uncaged much faster in our hands. Since we did not have caged carbamates available in our lab and use different LEDs and solvents as in the other reports using thio-DEACM we did not directly compare these results. We cite the papers for the interested reader (e.g. Chem. - A Eur. J. 2013, 19, 17494–17507). As our intention was to compare DEACM and thio-DEACM caged phosphates, we did not chose to go into a more detailed discussion about other leaving groups.
3) Cartesian coordinates and thermochemistry data should be given in SI for all optimized structures.
The cartesian coordinates have been added to the SI as supplementary tables S1 to S3.
We thank the reviewer for this very positive evaluation of our manuscript and hope that the clarifications and additions now merit publication.
Reviewer 2 Report
The manuscript titled “Thio-coumarin Caged Nucleotides: Synthetic Access 3 and Their Photophysical Properties” is a very interesting and well-elaborated report about the use of 7-diethylamino-4-hydroxymethyl-thiocoumarin as photocage, which could be useful as cellular signaling.
The authors carried out the synthesis of some coumarin derivatives that were submitted to photophysical properties analysis and comparison. The results showed that the substitution from O to S in the coumarin scaffold had several advantages in uncaging experiments.
The manuscript still contains very good Supplementary data, which provides an important data bank for the readers interesting in the topic.
In my opinion, the manuscript deserves to be published.
Author Response
We thank the reviewer for this very positive evaluation of our manuscript.